# Is There a Benefit of Oxaliplatin in Combination with Neoadjuvant Chemoradiotherapy for Locally Advanced Rectal Cancer? An Updated Meta-Analysis

**DOI:** 10.3390/cancers13236035

**Published:** 2021-11-30

**Authors:** Gaëtan Des Guetz, Thierry Landre, Marc A. Bollet, Muriel Mathonnet, Laurent Quéro

**Affiliations:** 1Medical Oncology Department, Delafontaine Hospital, 93200 St Denis, France; 2Department of Surgery, Faculty of Medicine, University of Limoges, 87032 Limoges, France; muriel.mathonnet@unilim.fr; 3Unité de Coordination en Onco-Gériatrie, Hôpitaux Universitaires Paris Seine-St-Denis, AP-HP, 93270 Sevran, France; thierry.landre@aphp.fr; 4Centre de Radiothérapie Hartmann, 92300 Levallois-Perret, France; marc.bollet@horg.fr; 5Department of Surgery, University Hospital of Limoges, 87032 Limoges, France; 6INSERM U1160, Université de Paris, 75010 Paris, France; laurent.quero@aphp.fr; 7Radiation Oncology Department, Saint-Louis University Hospital, AP-HP, 75010 Paris, France

**Keywords:** meta-analysis, randomized, rectal, radiotherapy, oxaliplatin, neoadjuvant

## Abstract

**Simple Summary:**

Neoadjuvant fluoropyrimidine (5FU or capecitabine)-based chemoradiotherapy (CRT) has been considered the standard of care for locally advanced rectal cancer (LARC). Our Meta-analysis showed that the combining oxaliplatin with capecitabine or 5FU in preoperative chemoradiotherapy or perioperative chemotherapy seems beneficial significantly and improved DFS. It remains necessary to identify which patients benefit most from the addition of oxaliplatin.

**Abstract:**

Background: Neoadjuvant fluoropyrimidine (5FU or capecitabine)-based chemoradiotherapy (CRT) has been considered the standard of care for locally advanced rectal cancer (LARC). Whether addition of oxaliplatin (OXP) will further improve clinical outcomes is still unclear. Methods: To identify clinical trials combining oxaliplatin in preoperative CRT or perioperative chemotherapy for LARC published until March 2021, we searched PubMed and the Cochrane Library. We also searched for relevant ASCO conference abstracts. The primary endpoint was disease-free survival (DFS). Data were extracted from every study to perform a meta-analysis using Review Manager (version 5.3). Results: A total of seven randomized clinical trials (ACCORD-12, CARO-AIO-04, FOWARC, JIAO, NSABP, PETACC-6, and STAR-01) with 5782 stage II or III rectal cancer patients were analyzed, including 2727 patients with OXP + 5FU regimen and 3055 patients with 5FU alone. Compared with the 5FU alone group, the OXP + 5FU regimen improved DFS (HR = 0.90, 95% CI: 0.81–0.99, *p* = 0.03) and pathologic complete response (pCR) (OR = 1.21, 95% CI: 1.07–1.37, *p* = 0.002). Patients treated with the OXP + 5FU regimen had significantly less metastatic progression (OR = 0.79; 95% CI, 0.67 to 0.94; *p* = 0.007). Considering adverse events (AEs), there was more grade 3–4 diarrhea with OXP + 5FU (OR = 2.41, 95% CI: 1.74–3.32, *p* < 0.00001). However, there were no significant differences grade 3–4 hematologic AEs (OR = 1.16, 95% CI: 0.87–1.57, *p* = 0.31). Conclusions: Our meta-analysis with long-term results from the randomized studies showed a benefit of the addition of OXP + 5FU regiment in terms of DFS, metastatic progression, and pCR rate that did not translate to improved OS.

## 1. Introduction

Colorectal cancer (CRC) is the third most deadly and fourth most commonly diagnosed cancer in the world according to GLOBOCAN 2018 data. Nearly 2 million new cases and about 1 million deaths were observed in 2018 [1]. The treatment of locally advanced rectal cancer has long been based on surgery with total mesorectal excision following chemoradiotherapy [2].

This therapeutic strategy made it possible to reduce the rate of loco-regional recurrence without, unfortunately, reducing the rate of distant metastatic recurrence, which remained high at around 30% [2,3]. This lack of efficacy in distant metastatic control could be explained by an insufficient dose-intensity/activity of fluoropyrimidine-based chemotherapy administered concomitantly with radiotherapy to control the micrometastatic disease. Oxaliplatin has been shown to be effective in the treatment of metastatic colorectal cancer and also in the adjuvant treatment of colon cancer. Moreover, oxaliplatin can also act as a radiosensitizer agent [4].

Based on these promising data, several randomized trials have been conducted to evaluate the efficacy of adding oxaliplatin to fluoropyrimidine-based chemoradiotherapy. Because of the inconsistent results of these trials, several meta-analyses have been carried out, but they have not been conclusive [5,6,7,8,9,10].

Following the recent update of the FOWARC and PETACC-6 clinical trials, we conducted an updated meta-analysis. The primary endpoint of this meta-analysis was to evaluate the benefit of the addition of oxaliplatin-based chemotherapy to concurrent fluoropyrimidine-based chemoradiotherapy and its impact on disease-free survival.

## 2. Patients and Methods

### 2.1. Literature Search

PubMed and Cochrane databases were screened on 30 May 2021. MeSH terms were used throughout the search schemes, which were adjusted appropriately in various electronic records. We also manually searched the abstracts accepted for the following major academic conferences: ESMO (European Society for Medical Oncology), ESTRO (Euro-pean Society for Radiation Oncology), ASCO (American Society of Clinical Oncology), ASTRO (American Society for Radiation Oncology), and ESSO (European Society of Surgical Oncology) until May 2021.

### 2.2. Trial Selection

The methodological quality of the selected studies was evaluated by two authors (T.L. and L.Q.). The trial selection process is documented by a PRISMA flow diagram giving specific reasons for exclusion of studies at each stage (Figure A1). We restricted the search to RCTs comparing neoadjuvant chemoradiotherapy with oxaliplatin added to 5-fluorouracil or capecitabine. Exclusion criteria were non-phase III randomized studies, letters, comments, and editorials and publications for which the full text was irretrievable. In case of multiple publications on a single clinical trial, all publications were included, and the all results were used with priority given to those with the longest follow-up.

### 2.3. Outcome Measures

The main outcome was disease-free survival. Additional parameters included local recurrence, metastatic progression, pathological complete response, pathological complete resection, postoperative complication rate, toxicity, and overall survival.

### 2.4. Statistical Analysis

The analyses were conducted according to the Cochrane method for meta-analyses and computed with Review Manager software (RevMan version 5.3; Oxford, UK). Hazard ratios (HR) were pooled in meta-analyses by the inverse variance method. Risk ratios (RR) and 95% confidence intervals (CI) were calculated for count data. I^2^ and Chi^2^ tests were used to assess studies-shared heterogeneity. A fixed-effects model was used when between-study heterogeneity was weak. When heterogeneity was strong, a randomized model was used. All tests were bilateral, with *p* < 0.05 defining statistical significance.

## 3. Results

Our meta-analysis included seven trials, with a total of 5782 patients: STAR-01 trial [11], ACCORD12/0405 trial [12,13,14], NSABP R-04 trial [15,16], CAO/ARO/AIO-04 trial [17], LIAONING CANCER HOSPITAL (JIAO) trial [18], FOWARC trial [19], and PETACC-6 trial [20].

Patient characteristics, including age, gender, clinical T and N staging, and tumor location, were well balanced between groups (Table 1).

### 3.1. Disease-Free Survival

All seven trials reported disease-free survival (DFS) in a total of 5782 patients. DFS was statistically significantly improved by the addition of oxaliplatin in the meta-analysis (HR = 0.90, 95% CI: 0.81–0.99; *p* = 0.03, I^2^ = 0%) (Figure 1).

### 3.2. Overall Survival

Seven trials reported overall survival (OS) corresponding to 5782 patients. No statistically significant difference was observed for OS (HR = 0.9495% CI: 0.83–1.06; *p* = 0.53, I^2^ = 0%) (Figure 2).

### 3.3. Pathological Complete Response

Seven trials reported pathological complete response rate after neoadjuvant chemoradiotherapy corresponding to 5386 patients. The pathological complete response rate ranged from 13.4% to 27.5% in the oxaliplatin + 5FU group and from 11.3% to 19.4% in the 5FU only group, with a significantly higher rate in the oxaliplatin + 5FU group (17.9% vs. 14.7%, RR = 1.21, 95% CI: 1.07–1.37, *p* = 0.002, I^2^ = 14%) (Figure 3).

### 3.4. Local Recurrence

Four trials reported local recurrence rate corresponding to 3635 patients. The local recurrence rate ranged from 3.0% to 11.5% in the oxaliplatin group and from 4.7% to 12.1% in the fluoropyrimidine only group, without statistical significance (7.0% vs. 8.1%, RR = 0.86, 95% CI: 0.68–1.08, *p* = 0.19, I^2^ = 0%) (Figure 4).

### 3.5. Metastatic Progression

Metastatic progression rate was reported in three studies, corresponding to a total of 2040 patients. The metastatic progression rate ranged from 16.5% to 22.1% in the oxaliplatin + 5FU group and from 22.5% to 28.2% in the 5FU only group, with a significantly lower rate in the oxaliplatin + 5FU group (18.7% vs. 23.6%, RR = 0.79, 95% CI: 0.67–0.94, *p* = 0.007, I^2^ = 2%) (Figure 5).

### 3.6. R0 Resection

R0 resection rate was reported in six studies, corresponding to a total of 4097 patients. The R0 resection rate ranged from 86.3% to 97.1% in the oxaliplatin + 5FU group and from 87.3% to 95.2% in the 5FU only group, without statistical significance (92.5% vs. 91.9%, RR = 1.05, 95% CI: 0.66–1.67, *p* = 0.83, I^2^ = 69%) (Figure 6).

### 3.7. Toxicity

Severe toxicities (grade 3–4) were reported in six studies, corresponding to a total of 5125 patients. Severe toxicity rates ranged from 21.4% to 40.1% in the oxaliplatin + 5FU group and from 7.6% to 25.5% in the 5FU group, without statistical significance (30.7% vs. 17.7%, RR = 1.92, 95% CI: 1.40–2.64, *p* < 0.0001, I^2^ = 87%) (Figure 7).

### 3.8. Hematological Toxicity

Severe hematological toxicities (grade 3–4) were reported in four studies, corresponding to a total of 2350 patients. Severe hematological toxicity rates ranged from 4.8% to 20.6% in the oxaliplatin + 5FU group and from 2.9% to 14.6% in the 5FU group, without statistical significance (7.3% vs. 6.3%, RR = 1.16, 95% CI: 0.87–1.57, *p* = 0.31, I^2^ = 0%) (Figure 8).

### 3.9. Digestive Toxicity

Severe diarrhea toxicities (grade 3–4) were reported in seven studies, corresponding to a total of 5455 patients. Severe diarrhea toxicity rates ranged from 12.1% to 18.5% in the oxaliplatin + 5FU group and from 3.1% to 9.7% in the 5FU only group, with a significantly higher rate in the oxaliplatin + 5FU group (15.1% vs. 6.4%, RR = 2.41, 95% CI: 1.74–3.32, *p* < 0.00001, I^2^ = 68%) (Figure 9).

### 3.10. Postoperative Complications

Postoperative complication rates were reported in five studies, corresponding to a total of 4818 patients. Postoperative complication rates ranged from 22.1% to 43.7% in the fluoropyrimidine only group and from 23.7% to 47.0% in the oxaliplatin group, without statistical significance (3.8% vs. 3.6%, RR = 1.05, 95% CI: 0.98–1.13, P = 0.15, I^2^ = 0%) (Figure 10).

### 3.11. Permanent Stoma

We did not find any significative difference between the two groups regarding permanent stoma incidence rate: RR = 1.01 (0.92–1.12).

### 3.12. Death within 60 Days

We did not find any significative difference between the two groups regarding death incidence within 60 days after surgery: RR = 0.83 (0.35–2.00).

## 4. Discussion

We performed a meta-analysis evaluating the effect of preoperative radiotherapy combined with fluoropyrimidine (capecitabine or 5FU) with or without oxaliplatin. As the aim of our study was to assess the radiosensitizing effect of oxaliplatin, we excluded the ADORE study from our meta-analysis, as it mainly assessed the benefit of postoperative treatment [21]. Indeed, the benefit of postoperative chemotherapy in rectal cancer is unclear despite no decrease in dose intensity of oxaliplatin in postoperative regimens as compared with colon cancer patients [22]. Conversely, oxaliplatin in the neoadjuvant setting has a growing interest in alternative approaches with less morbidity, including the organ-preserving watch-and-wait strategy, in which surgery is omitted in patients who have achieved a clinical complete response [23].

One of the limitations of the meta-analysis is that not only the doses/schedules of oxaliplatin differed between the trials but also the doses/schedules of 5FU/capecitabine differed between the trials and also within the same trial (CAO/ARO AIO-04). This could explain why some trials were significantly positive and others were not.

In the CAO/ARO/AIO-04 trial, patients received oxaliplatin not only during neoadjuvant chemoradiotherapy but also during adjuvant chemotherapy, and therefore patients received a higher cumulative total dose of oxaliplatin than in the other studies (1000 vs. 250–360 mg/m^2^). In this trial, pathological complete response (pCR) rate and 3-year DFS were improved. Moreover, in a retrospective study, Chang et al. showed that cumulative oxaliplatin dose (COD) <460 mg/m^2^ was an independent predictor of poorer overall metastasis-free and disease-free survival. However, a COD 460 mg/m^2^ increased the incidence of acute toxicities from 38.4% to 70.8% (*p* < 0.001) [24].

The primary endpoint for most studies analyzed in our meta-analysis was DFS (Jiao, FOWARC, CAO/ARO/AIO-04, PETACC6). It was OS in STAR-01 and Jiao, local–regional tumor control in the NSABP study, and pathologic complete response (ypCR) in ACCORD 12 (Table 2). In our meta-analysis, we found that the addition of oxaliplatin to radiotherapy increased metastasis-free survival and pathological complete response rate. Zheng’s meta-analysis of eight studies found an additional benefit in terms of local relapse [6].

### 4.1. Comparison with Others Meta-Analyses

Several meta-analyses have been previously published between 2013 and 2018 (Table 3). These studies addressed the same question of the benefit of adding oxaliplatin to the standard treatment of preoperative radiotherapy combined with a fluoropyrimidine.

The short follow-up of previously published meta-analyses could preclude the translation of improved DFS on OS. The latest meta-analysis was published more than 3 years ago, our meta-analysis incorporated the latest update of the FOWARC study results published in 2019 and the PETACC-6 study results published in 2021, with a median follow-up of 68 months vs. 31 months in the previous publication in 2013. One by one, the studies included in our meta-analysis did not find a statistically significant benefit in terms of DFS. However, our meta-analysis found a significant benefit in terms of DFS. This could be explained by the updating of data from the FOWARC study.

Despite the lasting benefit in DFS, our meta-analysis did not show a benefit of the addition of oxaliplatin in terms of overall survival. The main weakness of our study is that our meta-analysis was performed on published data and not on individual data. However, all selected studies were phase 3 studies conducted by cooperating clinical research groups producing reliable data.

### 4.2. Toxicity

Similar to other meta-analyses, we found an increase in toxicity with the addition of oxaliplatin.

Oxaliplatin increased hematological and gastrointestinal toxicity in comparison with fluoropyrimidine-based chemotherapy alone. These toxicities are manageable but require careful monitoring, especially diarrhea, which can be responsible for sepsis or dehydration. In clinical practice, this toxicity does not appear to be a definitive obstacle to the use of oxaliplatin in combination with radiotherapy.

### 4.3. Perspectives

Oxaliplatin is a major drug in digestive oncology. It is active in many cancers such as stomach, esophagus, pancreatic, and colon/rectal cancers. Given the effectiveness of fluoropyrimidine-based preoperative chemoradiotherapy in local control, the main criterion in improving the treatment of locally advanced rectal cancer should be distant control, as metastases still occur in about 30% of patients. The other improvement to be made would be to increase the rate of pathological complete response of the rectal tumor in order to avoid surgery and thus the risk of developing LARS syndrome or undergoing definitive colostomy, as could happen in abdominal-perineal amputation for very low rectal cancers.

The improvement in progression-free survival and pathological complete response rate observed in our meta-analysis with the concomitant addition of oxaliplatin to standard chemoradiotherapy must be weighed against the significant improvement in metastasis-free survival and histological complete response rate without redhibitory toxicity obtained with total neoadjuvant therapy combining induction FOLFIRINOX chemotherapy followed by chemoradiotherapy, as in the PRODIGE 23 randomized trial or with short course radiotherapy followed by oxaliplatin-based consolidation chemotherapy, as in the RAPIDO randomized trial recently published [25,26].

The administration of an optimal chemotherapy preoperatively and sequentially to the concomitant (chemo)-radiotherapy has made it possible to decrease the rate of metastatic progression from 30% to around 20% and to achieve a rate of complete pathological response around 30%, with very acceptable tolerance.

In accordance with these data, the total neoadjuvant therapy (TNT) approach could become one of the standard treatments for locally advanced rectal cancer. The question of the feasibility of a TNT associated with concomitant chemoradiotherapy potentiated by oxaliplatin remains, especially given radiotherapy technical progress, such as VMAT that decreases intestinal toxicity [27]. This approach could be interesting to evaluate in the context of a watch-and-wait organ conservation strategy.

## 5. Conclusions

Our meta-analysis with long-term results from the randomized studies showed a benefit of the addition of OXP + 5FU regiment in terms of DFS, metastatic progression, and pCR rate that did not translate to improved OS. It remains necessary to identify which patients benefit most from the addition of oxaliplatin.

## Figures and Tables

**Figure 1 cancers-13-06035-f001:**
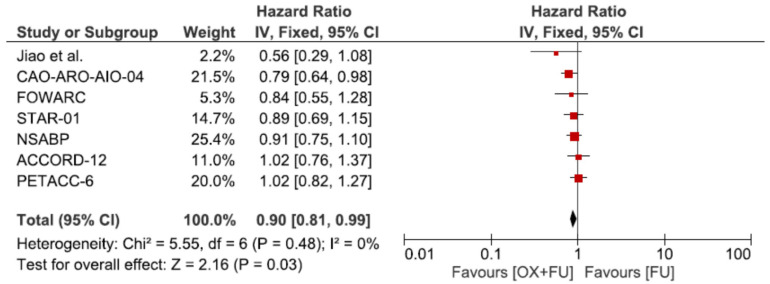
Forest plot of hazard ratio for disease free survival.

**Figure 2 cancers-13-06035-f002:**
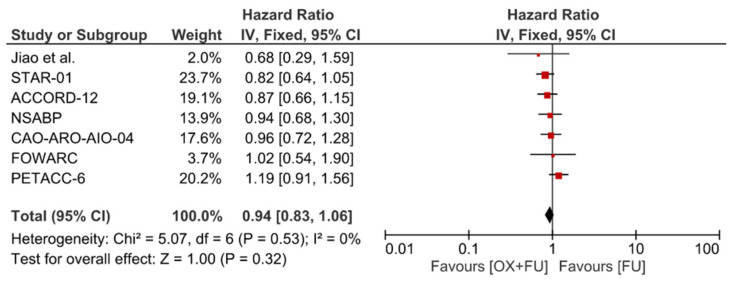
Forest plot of hazard ratio for overall survival.

**Figure 3 cancers-13-06035-f003:**
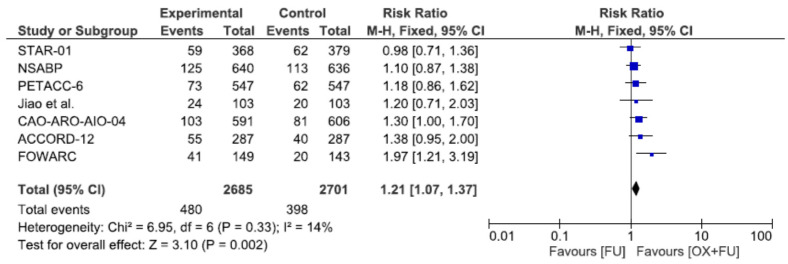
Forest plot of risk ratio for pathological complete response.

**Figure 4 cancers-13-06035-f004:**
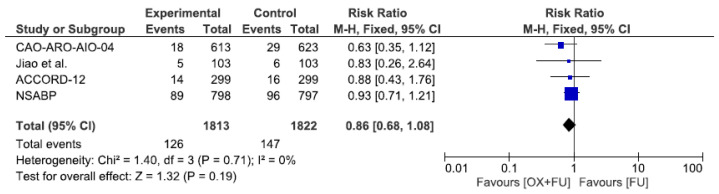
Forest plot of risk ratio for local recurrence.

**Figure 5 cancers-13-06035-f005:**
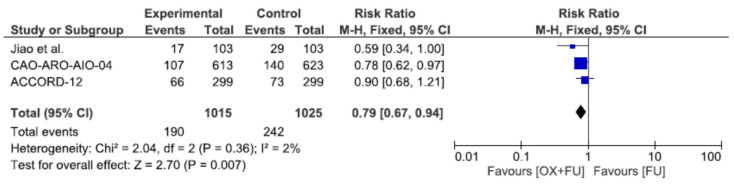
Forest plot of risk ratio for metastatic progression.

**Figure 6 cancers-13-06035-f006:**
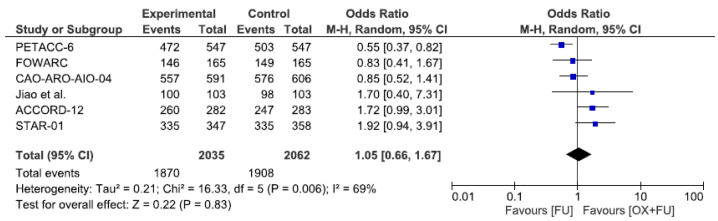
Forest plot of odds ratio for R0 Resection.

**Figure 7 cancers-13-06035-f007:**
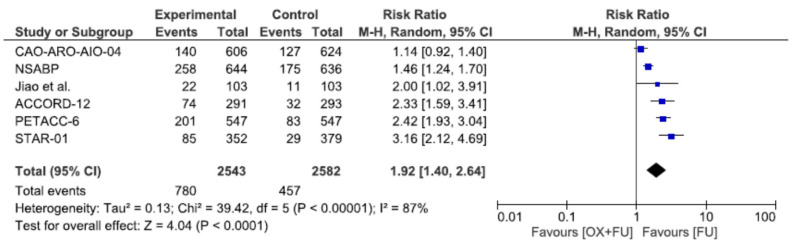
Forest plot of risk ratio for grade 3 or 4 toxicity.

**Figure 8 cancers-13-06035-f008:**
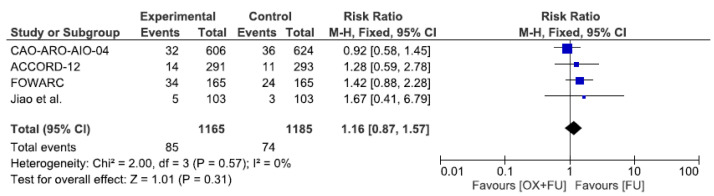
Forest plot of risk ratio for hematological toxicity.

**Figure 9 cancers-13-06035-f009:**
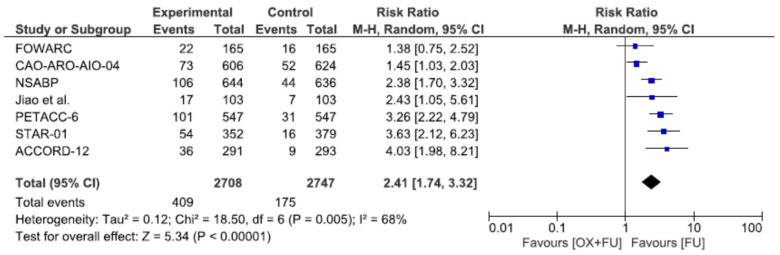
Forest plot of risk ratio for digestive toxicity.

**Figure 10 cancers-13-06035-f010:**
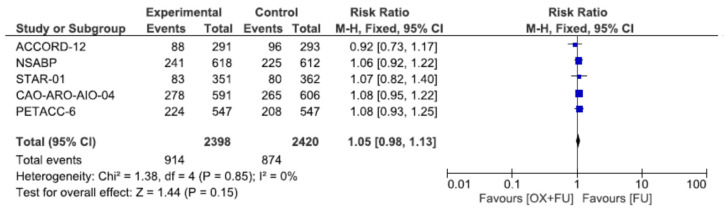
Forest plot of risk ratio for postoperative complications.

**Table 1 cancers-13-06035-t001:** Characteristics of included studies.

Study	Year	Country	Patients (n)	Radiotherapy	Concurrent Chemotherapy	Adjuvant Chemotherapy	Median F/U
NSABP-R04	2015	USA	Control = 949 Exp = 659	45 Gy or 50.4 Gy or 55.8 Gy (5 × 1.8 Gy/w)	Control: 5FU 225 mg/m^2^/d or CaP 825 mg/m^2^ × 2/d Exp: 5FU 225 mg/m^2^/d or CaP 825 mg/m^2^ × 2/d + oxaliplatin 50 mg/m^2^/w	NR	NR
STAR-01	2016	Italy	Control = 379 Exp = 368	50.4 Gy (5 × 1.8 Gy/w)	Control: 5FU 225 mg/m^2^/d Exp: 5FU 225 mg/m^2^/d + oxaliplatin 60 mg/m^2^/w × 6w	5FU-based	105.6 months
FOWARC	2019	China	Control = 165 Exp = 165	46.0 Gy (5 × 2Gy/w) or 50.4 Gy (5 × 1.8 Gy/w)	Control: leucovorin 400 mg/m^2^ 5FU bolus 400 mg/m^2^ + 5FU 2400 mg/m^2^ d1–d2/2w Exp: leucovorin 400 mg/m^2^ 5FU bolus 400 mg/m^2^ + 5FU 2400 mg/m^2^ d1–d2/2w + oxaliplatin 85 mg/m^2^/2w	mFOLFOX6 × 6 cycles	45.2 months
ACCORD-12	2017	France	Control = 293 Exp = 291	Control: 45 Gy (5 × 1.8 Gy/w) Exp: 50 Gy (5 × 2 Gy/w)	Control: CaP 800 mg/m^2^ × 2/d Exp: CaP 800 mg/m^2^ × 2/d + oxaliplatin 50 mg/m^2^/w	NR	60.2 months
CAO/ARO/AIO-04	2015	Germany	Control = 623 Exp = 613	50.4 Gy (5 × 1.8 Gy/w)	Control: 5FU 1000 mg/m^2^/d d1–d5 and d29–d33 Exp: 5FU 250 mg/m^2^/d d1–d14 and d22–d35 + oxaliplatin 50 mg/m^2^/d d1, d8, d22, d29	Control: 5FU bolus 500 mg/m^2^ d 1–d5 (× 4 cycles) Exp: oxaliplatin 100 mg/m^2^/d d1 and d15) + leucovorin 400 mg/m^2^/d d1 and d15) + 5FU 2400 mg/m^2^ d1–d2 and d15–d16	50 months
PETACC-6	2021	Europe	Control = 543 Exp = 528	45 Gy or 50.4 Gy (5 × 1.8 Gy/w)	Control: CaP 825 mg/m^2^ × 2/d d1–d33 Exp: CaP 825 mg/m^2^ × 2/d d1-d33 + oxaliplatin 50 mg/m^2^/d d1, d8, d15, d22 and d29	Control: CaP 1000 mg/m^2^ × 2/d d1–15 × 6 cycles Exp: CaP 1000 mg/m^2^ × 2/d d1–15 +oxaliplatin 130 mg/m^2^/d d1 × 6 cycles	68 months
Jiao et al. [18]	2015	China	Control = 103 Exp = 103	50 Gy (5 × 2 Gy/w)	Control: CaP 800 mg/m^2^ × 2/d d1–d14 and d22–d25 Exp: CaP 800 mg/m^2^ × 2/d d1–d14 and d22–d25 + oxaliplatin 60 mg/m^2^/d d1, d8, d22 and d29	5FU bolus 400 mg/m^2^/d. + 5FU 2400 mg/m^2^ d1–d2 + oxaliplatin 85 mg/m^2^/d + leucovorin 400 mg/m^2^ × 6–8 cycles	48.7 months
Total			Control = 3055 Exp = 2727				

Exp = Experimental; CaP = Capecitabine; d = day; w = week; Gy = Gray; NR = Not reported.

**Table 2 cancers-13-06035-t002:** Study-reported endpoints.

Study	OS	DFS	pCR	Local–Regional Recurrence	Metastatic Progression	R0	Toxicity
NSABP-R04	•	•	•	•*	•	•	•
STAR-01	•*	•	•	•	•	•	•
FOWARC	•	•*	•	•	•	•	•
ACCORD-12	•	•	•*	•	•	•	•
CAO/ARO/AIO-04	•	•*	•	•	•	•	•
PETACC-6	•	•*	•	•	•	•	•
Jiao et al. [18]	•*	•*	•	•	•	•	•

* primary endpoint; OS: overall survival; DFS: disease-free survival; pCR: pathological complete response; R0: negative resection margins. **•** endpoint evaluated; **•** endpoint not evaluated.

**Table 3 cancers-13-06035-t003:** Previous meta-analysis results.

Meta-Analysis	Years	Studies	n	OS	DFS	MFS	pCR	R0	Local Failure	Colostomy	Toxicity
Current study	2021	7	5782	-	+	+	+	-	-	-	+
Huttner	2018	5	5599	-	-	+	+	NR	-	NR	+
De Felice	2017	4	3310	-	-	+	NR	NR	-	NR	NR
Thavaneswaran	2017	7	4444	-	+	+	+	-	+		+
Zheng	2017	8	5597	-	+	+	+		+	-	+
Fu	2017	8	6103	-	+(3y) −(5y)	NR	+	NR	NR	-	+
Yang	2016	7	5415	NR	+	+	+	-	-	NR	+
Zhao	2016	4	2793	-	+	NR	NR	NR	NR	NR	+
Resende	2015	4	3875	-	-	-	+	-	-	-	+
An	2013	4	3863	NR	NR	+	+	NR	NR	-	+

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
