# Peer review of "Is There a Benefit of Oxaliplatin in Combination with Neoadjuvant Chemoradiotherapy for Locally Advanced Rectal Cancer? An Updated Meta-Analysis"

_cancers, 2021, doi:10.3390/cancers13236035_

Round 1

Reviewer 1 Report

Interesting topic! I think that add new knowledge to the topic!

Author Response

We sincerely thank reviewer number 1 for their appreciation.

Reviewer 2 Report

This paper is of interest. The updated meta-analysis focuses on DFS and OS benefit of oxaliplatin when added to CT-RT.

The main question addressed by the research is the role of oxaliplatin in addition to fluoropirimidines ad RT in the neoadjuvant setting of rectal cancer.

The topic is original, this is an open question.

I would not consider any changes in the methodology.

Conclusions are consistent with the evidence and arguments presented.

References are appropriate.

Tables and figures are clear and acceptable in format.

The meta-analysis is an update of previous works, including the latest research in the field.

I have no concerns, the paper deserves publication in this form.

Author Response

We warmly thank the reviewer number 2 for their analysis and positive comments.

Reviewer 3 Report

Thank you for the opportunity to review the manuscript.

This is an interesting article that presents the results of a meta-analysis of trials investigating the neoadjuvant treatment for locally advanced rectal cancer.

The aim of the study is clear and the research design with methodology looks proper.

In the introduction or discussion I recommend you to consider mentioning the fact that the benefit of postoperative chemotherapy in rectal cancer is unclear despite no decrease in dose intensity of oxaliplatin in postoperative regimens as compared with colon cancer patients (doi:10.1159/000501341); thus intensification of preoperative treatment may be a better strategy.

Moreover, consider discussing another interesting approach of intensifying preoperative treatment and then watch-and-wait strategy (see 10.3748/wjg.v26.i29.4218).

Author Response

We thank reviewer number 3 for his comments. Now we include in the discussion the 2 suggested proposals as well as their references.

Reviewer 4 Report

Manuscript entitled "Is there a benefit of oxaliplatin in combination with neoadjuvant chemoradiotherapy for locally advanced rectal cancer ? An updated meta-analysis"

This work measureed long-term results from the randomized studies showed a benefit of the addition of OXP + 5FU regiment in terms of DFS, metastatic progression and pCR rate that did not translate in improved OS. 

The major concerns are:

  1. This topic is not attractive.
  2. This work provides very limited if any new insights.
  3. This work is not so relevant to cutting-edge therapeutics.

Accordingly, this work should not be accepted by Cancers.

Author Response

We thank reviewer number 4 for his response. However we do not agree with his conclusions.

  1. This topic is a center of major interest given the frequency of rectal cancer.
  2. The place of oxaliplatin in the treatment strategy is still controversial, which justifies the interest of meta analyzes.
  3. The fact of providing new data seems important to us to improve the care of patients.

Round 2

Reviewer 4 Report

Apologize but there is no improvement in the revision.